# Single-molecule fluorimetry and gating currents inspire an improved optical voltage indicator

Jeremy S Treger[1†], Michael F Priest[2†], Francisco Bezanilla[1,2]*

[1]Department of Biochemistry and Molecular Biology, University of Chicago, Chicago, United States; [2]Committee on Neurobiology, University of Chicago, Chicago, United States

**Abstract** Voltage-sensing domains (VSDs) underlie the movement of voltage-gated ion channels, as well as the voltage-sensitive fluorescent responses observed from a common class of genetically encoded voltage indicators (GEVIs). Despite the widespread use and potential utility of these GEVIs, the biophysical underpinnings of the relationship between VSD movement and fluorophore response remain unclear. We investigated the recently developed GEVI ArcLight, and its close variant Arclight', at both the single-molecule and macroscopic levels to better understand their characteristics and mechanisms of activity. These studies revealed a number of previously unobserved features of ArcLight's behavior, including millisecond-scale fluorescence fluctuations in single molecules as well as a previously unreported delay prior to macroscopic fluorescence onset. Finally, these mechanistic insights allowed us to improve the optical response of ArcLight to fast or repetitive pulses with the development of ArcLightning, a novel GEVI with improved kinetics.

*For correspondence: fbezanilla@
peds.bsd.uchicago.edu

[†]These authors contributed equally to this work

**Competing interests:** The authors declare that no competing interests exist.

**Published:** 19 Noveber 2015

## Introduction

Accurately measuring membrane potential is crucial for understanding the activity of excitable cells. Traditionally, these measurements have been performed using various configurations of electrodes, but these can present numerous challenges such as mechanical disruption of cells and tissues and an inability to measure from many sites simultaneously with high spatial resolution (*Salzberg, 1989*). It has been found that the fluorescence of many membrane-bound organic dyes changes as a function of membrane potential (*Tasaki et al., 1968*; *Cohen et al., 1974*; *Loew et al., 1979*; *Davila et al., 1973*). Accordingly, this fluorescence can serve as an optical reporter of transmembrane voltage and render electrodes unnecessary. While this solves many problems associated with the use of electrodes, these potentiometric dyes stain cell membranes indiscriminately, leading to the labeling of glia and other tissue as well as neurons (*Jin et al., 2010*).

Fortunately, recent advances in optogenetics have provided an alternative to these techniques through the use of genetically encoded fluorescent voltage indicators (GEVIs) (*Baker et al., 2008*; *Jin et al., 2010*). These fluorescent proteins are expressed under the control of a desired genetic promoter and thus can be targeted to a chosen cell type. Once translated, the GEVIs are trafficked to the cell membrane where they sense changes in membrane potential and transduce this into a change in their fluorescence, thereby providing an optical readout of cellular electrical excitability. Many different designs have been employed to engineer GEVIs (*Siegel and Isacoff, 1997*; *Sakai et al., 2001*; *Dimitrov et al., 2007*; *Kralj et al., 2011*; *Akemann et al., 2012*). To date, most of these consist of a protein voltage-sensing domain (VSD) from a voltage-sensitive ion channel or phosphatase fused to one or more green fluorescent protein (GFP) derivatives. However, it remains largely unclear how movement of a VSD couples into changing fluorescence in an attached GFP,

**eLife digest** Nerve cells, or neurons, transmit information using changes in the voltage across their cell membranes. In the brain, these neurons work together in complex networks, and so understanding how the brain processes information will require neuroscientists to analyze voltage changes in many neurons at the same time. To achieve this, scientists have developed genetically-encoded voltage indicators (or GEVIs). These commonly feature a fluorescent protein attached to a voltage-sensitive protein; when the voltage-sensitive protein moves in response to changes in electrical activity, the amount of light emitted by the fluorescent protein also changes.

Treger, Priest and Bezanilla have now studied the characteristics of a popular GEVI called ArcLight by recording how fluorescence and voltage are related, both in single molecules and in groups of millions of molecules. This revealed that the fluorescence response of ArcLight does not occur instantly when a voltage change occurs. Instead the indicator fluoresces after a short delay. This delay corresponds with how quickly the voltage-sensitive protein responds. The fluorescence of a close relative of ArcLight also rapidly flickers, which deteriorates the signal quality.

Using this knowledge Treger, Priest and Bezanilla engineered the voltage-sensitive protein of ArcLight to develop a new variant of the indicator, named ArcLightning. Tests revealed that ArcLightning responds much faster than ArcLight to voltage changes in neurons, although the flicker of the fluorescent protein likely remains.

ArcLightning should prove to be a valuable tool for analyzing how neurons work together in living animals, but the flicker of the fluorescent protein suggests that there is further room for improvement. The rational design method used to develop ArcLightning could also be applied to improve other recently developed voltage indicators.

limiting the extent to which rational design can aid in the development and optimization of these GEVIs.

To help address this knowledge gap, we have conducted an extensive biophysical investigation of a voltage-sensitive phosphatase-based GEVI, ArcLight (*Jin et al., 2012*). We chose this molecule because it is one of the most promising GEVIs developed thus far with demonstrated utility in intact brain tissue (*Cao et al., 2013*). ArcLight is comprised of the VSD from the *Ciona intestinalis* voltage-sensor containing phosphatase (Ci-VSP) (*Murata et al., 2005*) fused to a variant of GFP. This voltage-sensing domain is highly similar to the VSD of voltage-gated ion channels; all contain a bundle of four transmembrane helices, with the fourth segment (S4) containing positively charged amino acids that behave as the primary sensors of voltage changes. The fluorescence of Arclight changes by more than 30% in response to a 100 mV change in membrane potential (*Jin et al., 2012*), an extremely large voltage-sensitive change compared to most other fluorescent proteins with S4-voltage sensors.

In this work, we first demonstrate that ArcLight fluorescence responds to motion of the attached voltage sensor and then investigate the characteristics of the fluorescence of ArcLight', a close homolog of ArcLight, at the single-molecule level. Although single GEVI molecules appeared to function normally, these traces unexpectedly displayed a significant degree of noise at millisecond timescales. Further investigations into this noise revealed that ArcLight' fluorophores possess intrinsic noise due to internal dynamics of the GFP moiety even in the absence of the voltage sensor, and that this noise appears to be a general feature of many GFP constructs including eGFP. Reducing or eliminating this fluctuation noise of the fluorophore could be a novel path towards improving this class of GEVIs. Finally, we investigated the behavior of ArcLight fluorescence at the macroscopic level and used this investigation to develop a novel voltage indicator with improved characteristics. ArcLight demonstrates distinct advantages over many other genetically-encoded voltage indicators, including large signal size relative to background, relatively low spectral bandwidth requirements compared to many FRET-based sensors (*Mishina et al., 2014*; *Akemann et al., 2012*; *2013*; *Tsutsui et al., 2013*), high quantum yield compared to archaerhodopsin based sensors (*Flytzanis et al., 2014*; *Gong et al., 2013*; *Hochbaum et al., 2014*; *Kralj et al., 2012*), and a demonstrated success in multiple biological preparations (*Jin et al., 2010*; *2012*; *Cao et al., 2013*; *Leyton-Mange et al., 2014*). However, ArcLight's ability to respond to rapid changes in membrane

potential is limited by its slow kinetics (*Jin et al., 2012*). To date, improvements in the kinetics of ArcLight have relied on switching residues from the *Ciona intestinalis* voltage-sensing domain (Ci-VSD) to residues from analogous voltage-sensitive phosphatases found in other species, especially *Gallus gallus* and *Danio rerio* (*Piao et al., 2015*; *Han et al., 2013*). Our mechanistic investigations into the link between ArcLight voltage sensor movement and fluorescence revealed an avenue of improvement for ArcLight as a voltage indicator. Using the cut-open oocyte voltage-clamp technique, we found that ArcLight fluorescence likely monitors a protein transition subsequent to gating charge movement, but that accelerating gating kinetics using a previously-reported mutation (*Lacroix and Bezanilla, 2012*) can nonetheless significantly speed up the fluorescence response. We termed this fast ArcLight derivative 'ArcLightning'. As expected from the accelerated kinetics of its Ci-VSD, ArcLightning expression in mammalian cells resulted in rapid and large voltage-sensitive fluorescence changes. This work shows that mutations discovered during biophysical studies of voltage sensor behavior can be used to rationally design improved GEVIs.

## Results

### ArcLight fluorescence changes are caused by voltage sensor movements

Our first goal was to validate that ArcLight fluorescence changes are driven by voltage sensor conformational changes, rather than by a different mechanism. While a direct response to membrane potential is unlikely as the fluorophore does not reside in the membrane electric field, it is conceivable that, for instance, the fluorophore is responding to localized pH changes near the membrane due to voltage-driven proton fluxes across the membrane. Using the cut-open oocyte voltage-clamp technique (*Stefani and Bezanilla, 1998*) we were able to simultaneously measure the gating current of ArcLight (*Figure 1A*) and its fluorescence (*Figure 1B*) from the same oocyte (*Cha and Bezanilla, 1998*). We then utilized the well-described mutations of the R217 residue of the Ci-VSP voltage sensor (*Dimitrov et al., 2007*; *Villalba-Galea et al., 2013*) to shift the voltage-dependence of the voltage sensor and observe the effects on the fluorescence. In ArcLight, this residue is mutated to a glutamine (R217Q), which places the midpoint of voltage sensor response at approximately -20 mV. Returning this residue to arginine (R217R) shifts the midpoint of gating charge movement to positive potentials while replacing the glutamine with glutamate (R217E) shifts the response to even more negative potentials (*Figure 1C*). Crucially, the fluorescence response mirrored these changes, indicating that ArcLight fluorescence signals are caused by voltage sensor movement rather than by direct influence of membrane potential or by voltage-driven ion concentration changes.

### Single ArcLight' molecules display millisecond-scale fluorescence fluctuations

Our next goal was to determine whether single GEVI molecules showed any unexpected behaviors that were being masked by ensemble averages. To do this, we visualized voltage-dependent fluorescence from single GEVI molecules using total internal reflection fluorescence microscopy (TIRFM). This technique presented an electrophysiological challenge, as TIRFM can only be performed on an oocyte once the vitelline membrane has been removed. The bare plasma membrane of the oocyte seals to the glass coverslip producing high access resistance to the bath electrodes and, consequently, a poor voltage clamp. Using an electrochromic voltage-sensitive small molecule dye (di-8-ANEPPS), we measured the speed of the voltage clamp on the underside of a peeled oocyte placed directly on a glass coverslip (*Figure 2—figure supplement 1*). As expected, the clamp was very slow due to the high access resistance. To remedy this situation, we coated glass coverslips with a polymer cushion. This cushion created a larger, conductive aqueous space between the oocyte and the glass that was thin enough to allow successful TIRFM. This was found to dramatically improve voltage clamp speed (*Figure 2—figure supplement 1*). This advance allowed us to successfully voltage clamp the molecules observable in TIRFM.

Having overcome this challenge, we turned our attention to observing single molecules of ArcLight' (see Methods section for details). Briefly, ArcLight' is a previously published variant of ArcLight (*Jin et al., 2012*) with two mutations in the GFP domain that seemed to improve its performance as a single-molecule fluorophore. Oocytes expressing low concentrations of ArcLight' had their vitelline

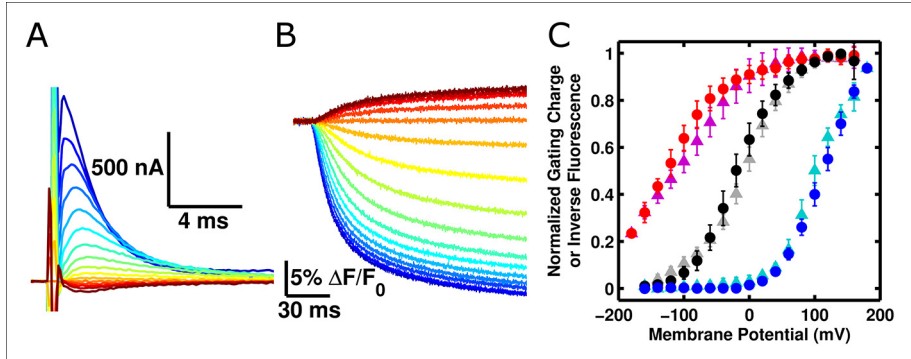

**Figure 1.** ArcLight fluorescence responds to voltage sensor movements. (A) Family of gating currents from an oocyte injected with ArcLight R217Q. The holding potential is -80 mV, and the pulses range from -160 mV (darkest red) to +140 mV (darkest blue) in 20 mV increments. (B) Fluorescence traces simultaneously acquired with gating currents in **A**. Colors indicate identical membrane potentials as in **A**. (C) Normalized gating-charge-versus-voltage (Q-V) curves (triangles; cyan – R217R, grey – R217Q, purple – R217E) and fluorescence-versus-voltage (F-V) curves (circles; blue – R217R, black – R217Q, red – R217E) for ArcLight. For clarity of comparison, the fluorescence is plotted as the change in fluorescence over the background fluorescence, multiplied by -1 (i.e., $-\Delta F/F_0$). Each Q-V curve was normalized by fitting a Boltzmann function to the data and scaling the limiting values of the function to 0 and 1. These scaling factors were then used to normalize each respective F-V curve. Error bars represent 95% confidence intervals of the mean. N = 5 for R217R, 7 for R217Q, and 6 for R217E.

membranes removed mechanically and were placed on polymer-coated coverslips and imaged with TIRFM. At the site of cRNA injection ArcLight' density appeared high and single molecules could typically not be resolved. However, imaging from fields further from the site of injection showed distinct, punctate sources of light consistent with small clusters and single molecules of ArcLight' (*Figure 2A*). As expected, many of these points showed clear changes in fluorescence in response to applied voltage changes, suggesting that they came from functional GEVI molecules (*Figure 2B*). To confirm that these single-molecules were behaving normally, traces from many different fluorescent spots were summed together (*Figure 2C*); these summed traces recapitulated the macroscopic response of ArcLight' to changes in membrane potential indicating that their function was uncompromised.

Although the single-molecules appeared to be operating normally, they consistently displayed an interesting phenomenon: a large degree of noise that did not appear to be shot noise. Whereas shot noise has no correlation from data point to data point, the single-molecule traces fluctuated between relatively distinct fluorescence levels, sometimes remaining at a single level for many milliseconds (*Figure 2B,C*). To our knowledge, there have been no prior reports of GEVIs displaying fluctuations on these millisecond timescales. Accordingly, we initially hypothesized that these fluctuations were reflective of voltage sensor conformational state changes. To test this, we bacterially expressed isolated ArcLight' GFP domains without attached voltage sensors and observed them as single molecules. To our surprise, these isolated GFP domains displayed roughly similar fluctuations to those observed in the full ArcLight molecules (*Figure 3A*), indicating that these fluctuations originate in the fluorophore itself. To confirm that this fluctuation noise was distinct from shot noise, we computed the autocorrelation functions for fluorescence traces from both an isolated ArcLight' GFP domain and from a region of background fluorescence from the same image (*Figure 3B,C*). As expected, the background fluorescence displayed no significant autocorrelation, suggesting that the noise in this trace is primarily due to shot noise or other white noise processes. By contrast, the ArcLight' fluorescence showed significant correlation over tens of milliseconds. Finally, we performed a similar experiment on wild-type eGFP and also observed significant fluctuation noise (*Figure 3D*). This result, consistent with the GFP photophysics literature, strongly suggests that this is not a pathology unique to the ArcLight' fluorophore, but rather is one that likely affects many current GEVIs (see Discussion). This observation may lead to a novel avenue for improvement of GEVI signals based on the attenuation or elimination of this noise.

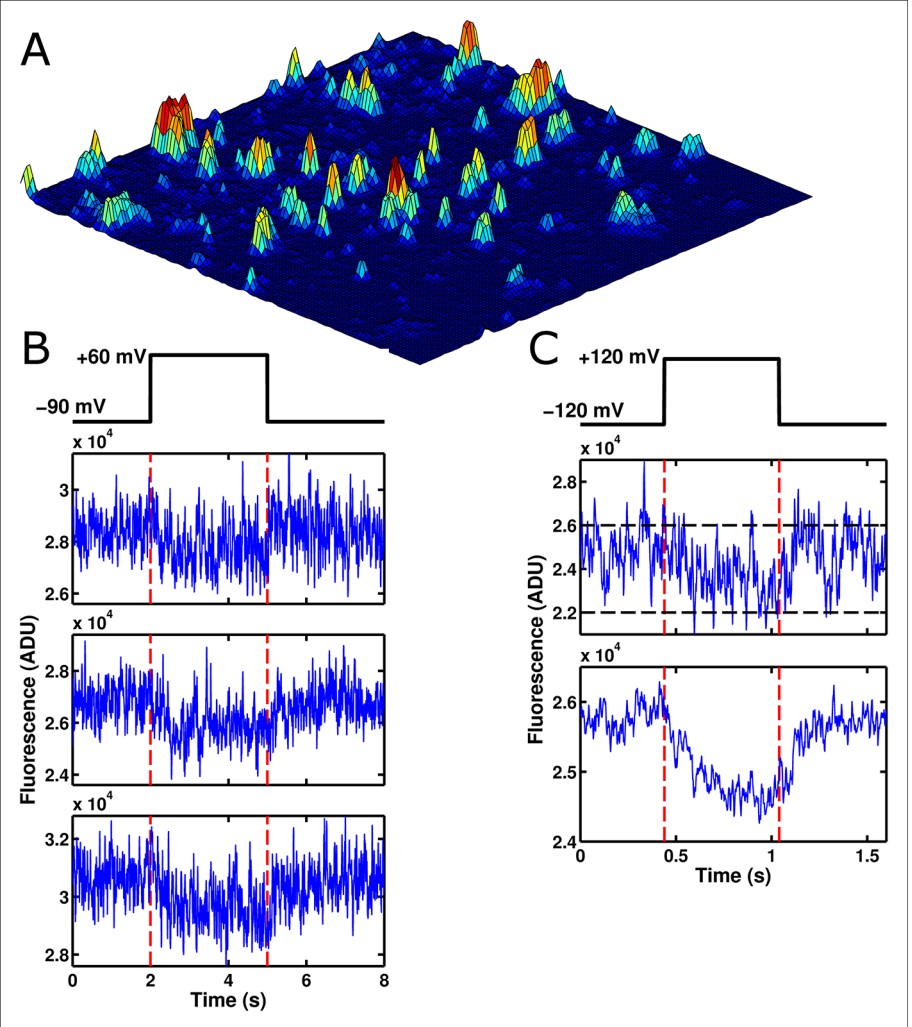

**Figure 2.** Voltage-sensitive fluorescence can be recorded from single ArcLight' molecules. (**A**) An image of the surface of an oocyte expressing ArcLight' in single-molecule concentrations. Square image field is 50 µm × 50 µm. Height in the z-axis corresponds to fluorescence intensity. This image is an average of 800 frames. Background correction has been performed using morphological filtering. (**B**) Three example traces are shown of single-molecule ArcLight' fluorescence in response to a three second depolarizing pulse from -90 mV to +60 mV. Traces were filtered at 40 Hz and had a linear baseline subtraction applied. (**C**) The upper of the two traces shows an example trace of single-molecule ArcLight' fluorescence from -120 mV to +120 mV for 600 ms (red dashed lines mark the beginning and end of the depolarizing pulse). The lower of the two traces is the average of 67 traces. This summation demonstrates that the single-molecule data here recapitulates the macroscopic voltage-sensitive fluorescence response of Arclight. Both traces were filtered at 100 Hz. A linear baseline subtraction was applied to the summed trace. ADU is analog-to-digital units, the output unit of the EMCCD camera.

The following figure supplements are available for Figure 2:

**Figure supplement 1.** A polymer cushion improves clamp speed under a peeled oocyte.

## ArcLight fluorescence changes are slower than gating currents and possess a lag

Having investigated the characteristics of single ArcLight fluorophores, we turned our attention towards understanding the macroscopic response of this molecule. As discussed previously, altering the set point of the voltage-dependence of the Ci-VSP voltage-sensing domain by mutating the R217 residue to either a Q or an E correspondingly alters the set point of the ArcLight fluorescence change (*Figure 1C*). Indeed, the change in ArcLight fluorescence at steady state was not statistically

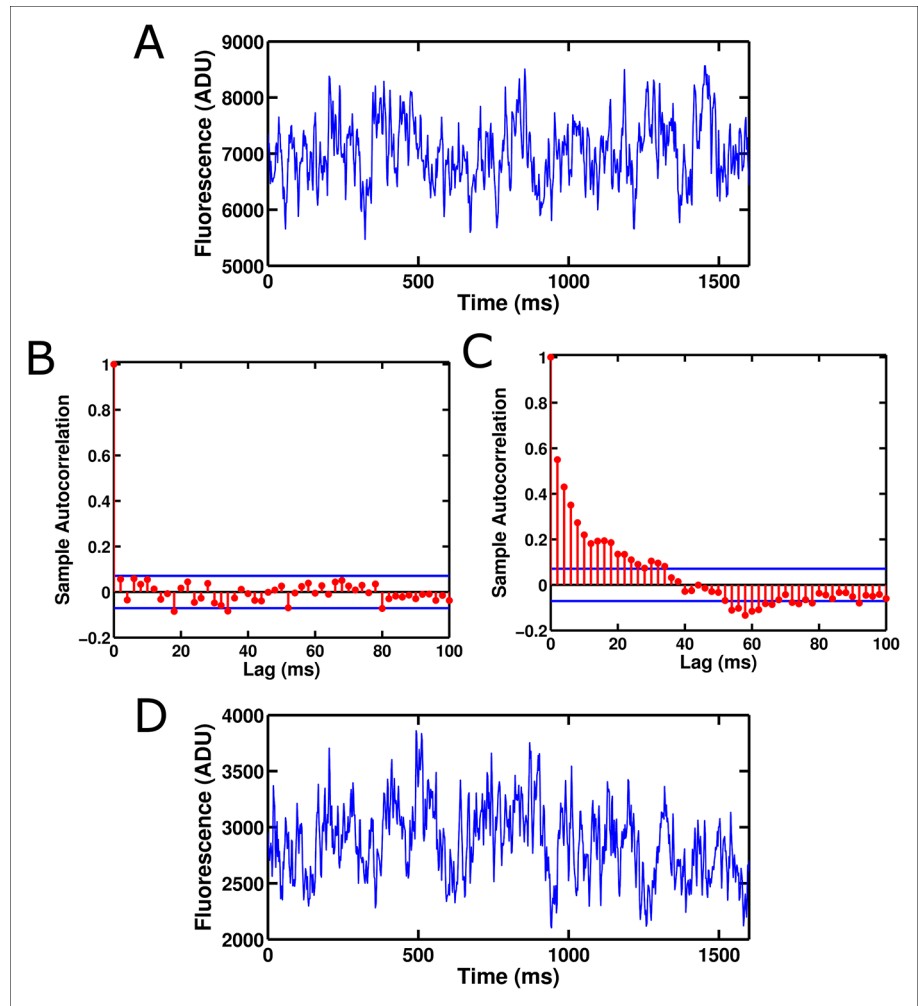

**Figure 3.** Fluorescence from an isolated ArcLight' GFP domain and eGFP show significant fluctuation noise. (**A**) Fluorescence from a single isolated GFP domain from ArcLight' with no attached voltage sensor. The fluorescence displays fluctuations between different levels and does not appear to be Poisson-distributed. The trace was filtered at 100 Hz. (**B**) Background fluorescence from a region of an image with no GFP molecule displays no significant autocorrelation at any non-zero lag. This suggests that most noise in this regime is due to white processes such as shot noise. (**C**) The fluorescence from a single isolated GFP domain of ArcLight' displays significant autocorrelation. This suggests that the fluorescence moves between distinct levels corresponding to different conformational, chemical, or electronic states of the GFP domain. Blue bars in **B** and **C** represent approximate 95% confidence intervals for a Gaussian-distributed white noise process. No filtering was applied to the data used for the autocorrelations. (**D**) Single molecules of wild-type eGFP also display fluctuation noise that appears qualitatively similar to that of ArcLight' GFP domains, suggesting that this noise may be present in many GFP derivatives. The trace was filtered at 100 Hz. ADU is analog-to-digital units, the output unit of the EMCCD camera.

significantly different from the degree of change in Ci-VSD conformation as measured by gating charge movement. We also tested whether ArcLight fluorescence kinetics are the same as the kinetics of the gating charge movement (see Methods for details). In both 'on' responses where the membrane was held at a negative potential and then pulsed to depolarized potentials (*Figure 4A*) and 'off' responses where the membrane was held at a positive potential and then pulsed to hyperpolarized potential (*Figure 4B*), we observed that ArcLight fluorescence kinetics are considerably slower than those of voltage sensor movement (*Figure 4C,D*). The fastest component of fluorescence during the 'on' response was approximately twice as slow as the gating current kinetics, and the 'off' fluorescence kinetics are more than ten times slower than voltage sensor gating. Indeed, neither

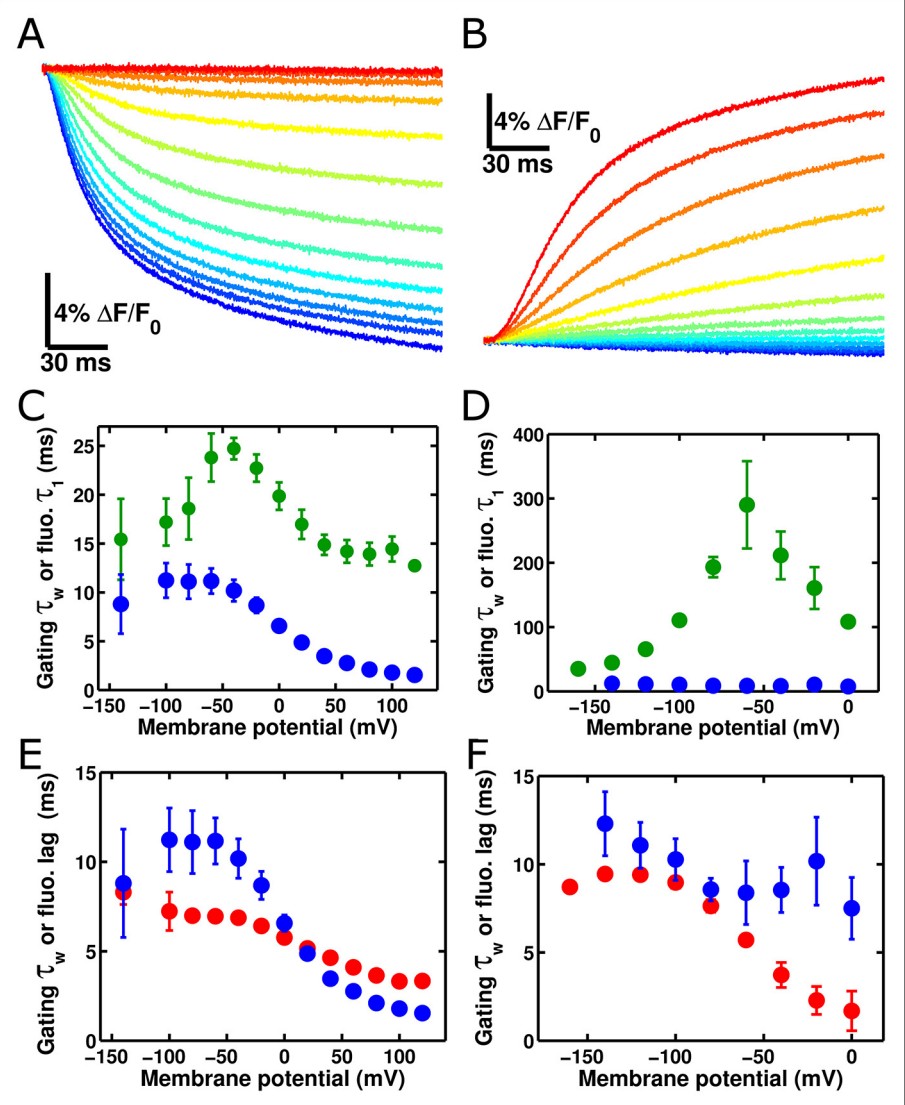

**Figure 4.** ArcLight fluorescence partially follows the kinetics of voltage sensor movement. (**A**) ArcLight fluorescence response to an 'on' pulse protocol with a holding potential of -120 mV with 200 ms pulses ranging from +120 mV to -140 mV by 20 mV intervals. (**B**) As in **A**, but in response to an 'off' pulse protocol pulsing from a holding potential of +40 mV to 200 ms pulses ranging from -160 mV to +40 mV by 20 mV intervals. (**C**) ArcLight on gating kinetics (blue) are faster than and do not correlate strongly with ArcLight fluorescence kinetics (green). Kinetics were obtained from gating currents and fluorescence changes recorded simultaneously from the same oocyte. (**D**) ArcLight off gating kinetics (blue) are much faster than and do not correlate with ArcLight fluorescence kinetics (green). (**E**) Upon depolarization, ArcLight fluorescence change shows a distinct lag (see ***Figure 4—figure supplement 1***). This lag is voltage-dependent, becoming shorter with more extreme changes in membrane potential. ArcLight gating current kinetics from the on pulse (blue) correlate quite well with the ArcLight fluorescence lag (red). (**F**) ArcLight gating current kinetics from the off pulse (blue) correlate with ArcLight fluorescence lag (red) better than they do with ArcLight fluorescence kinetics (shown in **D**, green). N = 4 for on pulse protocol data, 5 for off pulse protocol data.

The following figure supplements are available for Figure 4:

**Figure supplement 1.** ArcLight fluorescence changes are slower than integrated gating charge kinetics.

**Figure supplement 2.** The onset of ArcLight fluorescence change lags behind voltage change onset.

*Figure 4. continued on next page*

*Figure 4. Continued*

**Figure supplement 3.** The R217R and R217E ArcLight mutants show similar behaviors to R217Q, but with shifted voltage dependence.

component of fluorescence response appears to correlate well with any component of gating charge movement (*Figure 4—figure supplement 1*). Overlaying simultaneously acquired gating charge and fluorescence traces, along with an estimated measure of membrane potential, makes the slowed response of ArcLight fluorescence especially apparent (*Figure 4—figure supplement 1*). These observations raised the question of how gating charge movement is kinetically linked to ArcLight fluorescence. Interestingly, there is a distinct delay between the induction of a voltage change and the initiation of the resulting fluorescence change from ArcLight (*Figure 4—figure supplement 2*, *Figure 4E,F*). This delay shows a marked voltage-dependence and is present in both on and off fluorescence transitions. Both the delays and fluorescence kinetics show pronounced voltage dependencies, with $\tau$-V curves which are slowest at intermediate potentials and which speed up towards limiting values at extreme potentials. This pattern is reminiscent of gating current $\tau$-V curves, and suggests that voltage sensor movement is partially responsible for determining ArcLight fluorescence kinetics. In fact, the delays of the fluorescence response align quite well with the gating current kinetics (*Figure 4E*). In addition to wild-type ArcLight, we also investigated the kinetics of fluorescence response in relation to gating current kinetics for the R217R and R217E constructs (*Figure 4—figure supplement 3*). These two mutants generally behaved similarly to the R217Q construct with fluorescence lags of a comparable speed to VSD movement and a considerably slower fluorescence response.

## Accelerating ArcLight gating accelerates fluorescence responses, up to a limiting speed

To investigate these behaviors further, we tested whether altering the kinetics of Ci-VSD movement altered the kinetics of either the slow ArcLight fluorescence change or the lag before fluorescence change. By mutating the residue I126 in the S1 segment of the voltage-sensing domain of Ci-VSP to a phenylalanine, gating current activation kinetics were previously shown to be accelerated roughly forty-fold while deactivation currents were accelerated roughly eighty-fold (*Lacroix and Bezanilla, 2012*). We hypothesized that making the same mutation in ArcLight might speed up fluorescence responses in addition to gating currents. As the I126F mutation also shifted the Ci-VSP Q-V to more hyperpolarized potentials, the Q at position 217 in ArcLight was mutated back to R to keep the voltage-dependence at roughly physiological levels (*Figure 5A*). Since Ci-VSP R217R gating currents are about four-fold slower than R217Q gating currents (*Villalba-Galea et al., 2013*), we would expect that ArcLight I126F Q217R would display gating current kinetics about 10 times faster in response to depolarizing pulses and 20 times faster in response to hyperpolarizing pulses than wild-type ArcLight. In practice, we observe a roughly six to twelve-fold increase in speed (*Figure 5B,C Figure 5—figure supplement 1*). Accordingly, if ArcLight fluorescence kinetics track directly with the kinetics of its voltage sensing domain, the fluorescence changes of ArcLight I126F should be about 8 times faster than wild-type ArcLight during both depolarizations and hyperpolarizations. However, while I126F did accelerate the kinetics of ArcLight fluorescence, it did so in an interesting manner (*Figure 5D,E*). Specifically, while wild-type ArcLight fluorescence kinetics show a pronounced dependence on voltage, I126F kinetics are quite flat over most of the physiological voltage range. Furthermore, as the wild-type kinetics speed up towards extreme membrane potentials, they typically seem to approach the near-constant value of I126F. Interestingly, the lags before fluorescence movement seem to show a very similar pattern (*Figure 5F,G*). An additional feature is the remarkable acceleration of I126F fluorescence kinetics upon hyperpolarization compared to those from ArcLight (*Figure 5E*). These observations provide some insight into the biophysical basis of ArcLight fluorescence sensing (see Discussion). Due to its improved kinetic response versus wild-type ArcLight, we have termed the I126F Q217R mutant 'ArcLightning'.

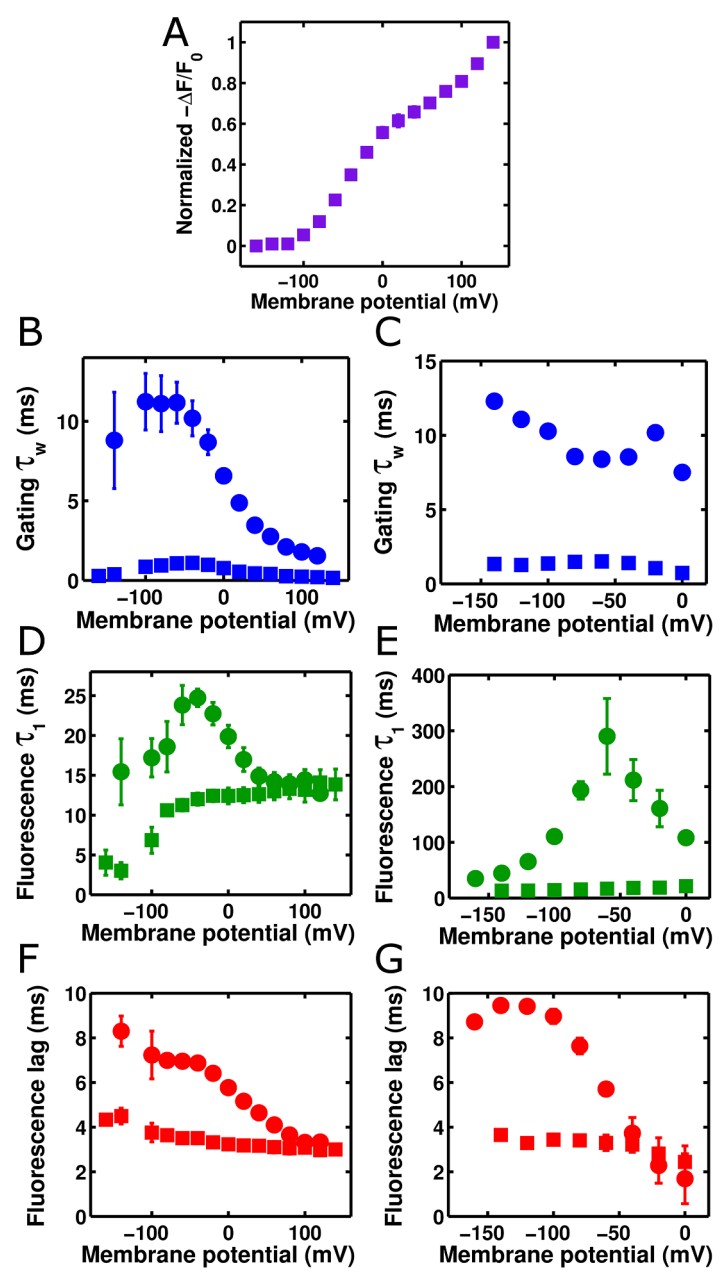

**Figure 5.** Accelerated gating kinetics speed up fluorescence, up to a point. (**A**) The voltage-dependence of fluorescence change of ArcLight I126F Q217R occurs over a physiological voltage range, as taken from an on protocol identical to that in **Figure 4**. (**B**) The gating current kinetics of ArcLight I126F Q217R (squares) obtained from the on pulse protocol are faster than those measured from ArcLight (circles). (**C**) The gating current kinetics of ArcLight I126F Q217R (squares) obtained from an off pulse protocol identical to that in **Figure 4** are faster than those measured from ArcLight (circles). (**D**) The fluorescence kinetics of ArcLight I126F Q217R (squares) in response to a depolarizing pulse are generally faster than those measured from ArcLight (circles). A notable exception is at more positive potentials, e.g. +80 mV, where the kinetics of the two constructs are essentially equal. (**E**) The fluorescence kinetics of ArcLight I126F Q217R (squares) obtained in response to a hyperpolarizing pulse are much faster than those measured from ArcLight (circles). (**F**) The fluorescence lag of ArcLight I126F Q217R (squares) during on pulses is briefer than the lag seen in ArcLight's fluorescence (circles). (**G**) As in **F**, but the fluorescence lag upon an off pulse. On and off protocols were as in **Figure 4**, and n = 5 for all ArcLight I216F Q217R data.

*Figure 5. continued on next page*

*Figure 5. Continued*

The following figure supplements are available for Figure 5:

**Figure supplement 1.** Substitution of I126F and Q217R into ArcLight induces a roughly constant acceleration of gating current.

## ArcLightning is a novel GEVI with improved kinetic response

Given its faster response relative to ArcLight, we suspected that ArcLightning might be useful as an improved GEVI. When expressed in HEK cells and measured at 19°C, ArcLightning showed signals which were substantially faster than those of ArcLight, particularly when returning to hyperpolarized potentials (*Figure 6A*). The acceleration of kinetics observed from ArcLight to ArcLightning in oocytes is recapitulated in mammalian cells (*Figure 6—figure supplement 1*). The fast component of ArcLightning fluorescence is faster than the fast component of ArcLight fluorescence at physiological membrane potentials, and the kinetics of the two constructs converge to similar values at extreme potentials where the ArcLight gating is faster. Furthermore, over much of the physiological voltage range, the fractional amplitude of the fast component is larger in ArcLightning than in ArcLight. The increased speed of return from a depolarizing pulse is particularly noticeable in the case of repetitive stimuli (*Figure 6B*). In this case, ArcLightning is able to return to baseline fluorescence in between consecutive voltage steps, providing an accurate readout of the size of each step. ArcLightning also retains its improved kinetics at 35°C. In response to trains of short pulses with durations emulating action potentials at 67 Hz, ArcLightning is still able to recover to baseline in between pulses (*Figure 6C*). ArcLight, by contrast, does not recover in time and thus shows an apparent increase in magnitude of each successive voltage pulse, despite them all being of equivalent magnitude. Although the larger net signal size of ArcLight may remain advantageous in some situations, the faster kinetics of ArcLightning should help in the resolution and analysis of many fast phenomena.

## Discussion

Here we report three advances in our understanding of GEVIs in general and ArcLight in particular. First, we show that the ArcLight' GFP domain possesses considerable intrinsic fluctuation noise on the millisecond timescale, and that this noise is also present in wild-type eGFP. Second, we demonstrate that for ArcLight, the transitions of the voltage-sensing domain align with the transitions of the fluorescence signal, although the fluorescence changes are significantly slower than the voltage sensor changes. Finally, we used a point mutation discovered during biophysical investigations of Ci-VSP to design ArcLightning, a novel version of ArcLight with accelerated movements of the voltage-sensing domain and improved fluorescence response times.

Our single-molecule analysis of ArcLight' revealed a previously unrecognized behavior of the GEVI: an inherent noise, or fluctuation in the fluorescence signal. This noise appears to not be Poisson-distributed, and thus is not likely simply attributable to shot noise from a single fluorophore. Furthermore, we observed similar fluctuations in single molecules of eGFP. There have been prior reports of fluorescence fluctuations in various GFP derivatives which generally fall into two categories: millisecond and microsecond-scale fluctuations detected with fluorescence correlation spectroscopy (*Hess et al., 2004*; *Haupts et al., 1998*; *Bosisio et al., 2008*), and much slower blinking detected in imaging studies of gel-immobilized fluorophores that displayed on- and off- dwell times that are typically hundreds of milliseconds and seconds, respectively (*Dickson et al., 1997*; *Peterman et al., 1999*; *Garcia-Parajo et al., 2000*); the latter of these modes of fluctuation appears to be quite different from what we observe. We have also, however, found one prior report of fluctuation in an immobilized GFP mutant on comparable timescales to what we observe (*Moerner et al., 1999*), further emphasizing that this noise may be a general feature of GFP derivatives. Crucially, however, this noise has to our knowledge never been verified or acknowledged in a GEVI. This is important as this noise adversely impacts the use of GEVIs in two ways. First, while the fluctuations themselves are not visible in macroscopic recordings due to the large number of fluorophore signals being averaged together, they very likely sum together in some manner to raise the noise level of

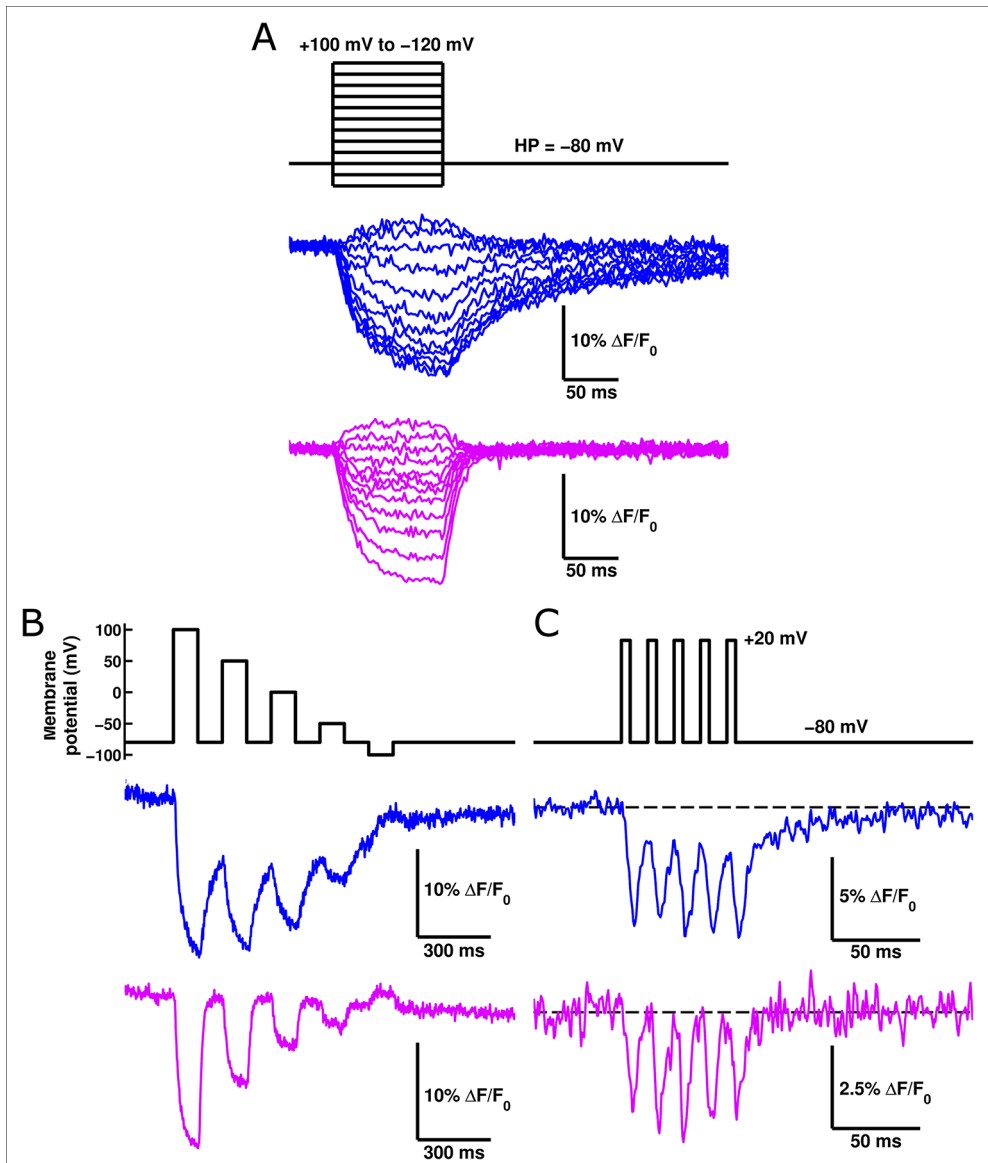

**Figure 6.** ArcLightning displays improved fluorescence response to repetitive pulses in mammalian cells. (**A**) ArcLightning (pink) expressed in HEK cells displays moderately faster kinetics in response to depolarization and much faster kinetics in response to hyperpolarization than ArcLight (blue) at 19°C. Notice that ArcLightning develops a slower component only at potentials exceeding +50 mV. (**B**) In response to a train of 100 ms pulses, separated by 100 ms of holding at -80 mV, ArcLightning (pink) displays an improved ability to measure discrete events compared to ArcLight (blue). All traces in **A** and **B** had no baseline subtraction or filtering. (**C**) At 35°C, both ArcLight (blue) and ArcLightning (pink) follow 5 ms, 100 mV pulses separated by 10 ms at the resting potential to mimic action potentials at approximately 67 Hz. ArcLightning provides the additional advantage of abolishing the sloping baseline of fluorescence that ArcLight produces in response to repetitive pulses as a result of its slow off kinetics. Both traces had a single-exponential baseline subtracted and were low-pass filtered at 300 Hz.

The following figure supplements are available for Figure 6:

**Figure supplement 1.** ArcLightning is faster than ArcLight in mammalian cells.

macroscopic recordings. Second, molecules in the dark state do not contribute useful 'signal' photons to help overcome 'noise' photons delivered by autofluorescence and other sources. Both of these effects likely serve to decrease the signal-to-noise ratio observed for a given set of excitation

parameters as compared to a non-fluctuating GEVI, thus limiting the size of phenomena that can be resolved. Accordingly, the reduction or elimination of this noise would likely be a significant benefit to GEVIs, thus providing a new direction for future GEVI optimization. In addition, prior work has shown that pulsed excitation light can significantly increase the amount of signal derived from fluorophores that visit dark states (*Mejía-Alvarez et al., 2003*; *Donnert et al., 2007*). By showing that the ArcLight' fluorophore makes frequent fluctuations, this work motivates investigations into whether an appropriately-modulated excitation can improve the signal from this and other GEVIs.

Despite its popularity as a GEVI, the mechanism by which ArcLight transduces the motion of the VSD into a change in the fluorescence of the attached eGFP remains largely unknown (*Han et al., 2013*; *Han et al., 2014*). However, the data presented here provides some insight into this process. The gating current kinetics of ArcLight look qualitatively similar to those from Ci-VSP R217Q (*Figure 1A*) with roughly equal kinetics and $V_{1/2}$ (11 ms versus 8 ms, -7 mV versus -16 mV) (*Figure 4C*, *Figure 1C*) (*Villalba-Galea et al., 2013*). This suggests that the attachment of the fluorescent protein is producing at most a small change in load on the voltage sensor movement, as compared to the natural load of the linker and the phosphatase in Ci-VSP R217Q. However, in ArcLight the fluorescence changes are substantially slower than the simultaneously measured gating currents (*Figure 4C*), and contain a measurable lag prior to onset (*Figure 4E*, *Figure 4—figure supplement 1*). This initially suggested a three-state minimal model of fluorescence with two state transitions: the first carrying gating charge movement and the second corresponding to fluorescence change (*Figure 7—figure supplement 1*). The lag would be accounted for by the time required for a significant population to reach the second state, while a smaller rate constant on the second transition than the first transition would cause the fluorescence response to be slower than gating charge movement.

To test this three-state model, we accelerated the gating charge movement of ArcLight with the I126F mutation. If the three-state model (*Figure 7—figure supplement 1*) was accurate, we would expect the lag preceding fluorescence change to decrease roughly in proportion to the quickening in gating current kinetics while maintaining a similar τ-V dependence. Instead, we observed that both the lag and the fluorescence kinetics become only weakly dependent on voltage. Furthermore, the value they take is approximately the same value that wild-type ArcLight reaches at extreme potentials when its voltage sensor is moving maximally fast. In particular, the lag preceding fluorescence does not speed up by nearly the same factor that the gating current speeds up. These observations are more consistent with a four-state minimal model (*Figure 7*). Here, as in the previous

$$S_0 \underset{\Delta Q}{\overset{k_1 = f(V)}{\rightleftharpoons}} S_1 \underset{Lag}{\overset{k_2 \neq f(V)}{\rightleftharpoons}} S_2 \underset{\Delta F}{\overset{k_3 \neq f(V)}{\rightleftharpoons}} S_3$$

**Figure 7.** minimal four-state model of ArcLight kinetics. The first transition carries the gating charge movement while the third transition carries the fluorescence change and the second transition carries no observable changes. The kinetics of the first transition depend strongly on voltage, while the kinetics of the second two are only weak functions of voltage. Thus, if $k_1 \gg k_2, k_3$, the second two transitions will become rate-limiting, and both the lag and fluorescence kinetics will appear nearly voltage-independent, as observed when gating is accelerated with the I126F mutation or in wild type ArcLight when gating becomes fast due to pulsing to extreme potentials. This model explains why wild-type ArcLight fluorescence kinetics and lags approach the same values as the I126F mutant at extreme potentials.

The following figure supplements are available for Figure 7:

**Figure supplement 1.** A minimal three-state model of ArcLight kinetics.

**Figure supplement 2.** ArcLight fluorescence may be influenced by the relaxed state.

model, the first transition carries the gating charge and possesses voltage-dependent kinetics. The second transition accounts for the lag and possesses kinetics which do not strongly depend on voltage. Finally, the third transition is responsible for the change in fluorescence and also has nearly voltage-independent kinetics. Thus, in wild-type ArcLight, the first transition contributes significantly to the overall observed time constants, causing the observed lags and fluorescence kinetics to behave as if they were voltage-dependent. In ArcLightning, however, the first transition is so fast that it does not contribute significantly and the next two transitions essentially become rate-limiting. This results in the observed values, which do not depend strongly on voltage.

Taken together, these data suggest that ArcLight fluorescence changes are slower than gating currents primarily due to an intrinsic delay in the fluorophore or linker, rather than the fluorescence accurately reporting on late VSD movements such as relaxation (*Villalba-Galea et al., 2008*). There are two pieces of evidence in favor of this intrinsic delay. First, ArcLight still shows fluorescence responses at extreme hyperpolarized potentials where the VSD is unlikely to enter the relaxed state. Second, speeding up VSD movement does not make the fluorescence response arbitrarily fast. Rather, there seem to be voltage-independent intrinsic delays in fluorescence response which become rate-limiting once VSD movement is fast enough (*Figure 5*). Thus, at least some component of the ArcLight response is likely due to VSD activation coupled with intrinsic fluorophore delay.

While ArcLight fluorescence does not seem to be a direct readout of VSD relaxation, the relaxed state may nonetheless play a role in the very slow response of fluorescence to hyperpolarizing pulses following a prolonged depolarization (*Figure 4D*). This is suggested by two observations: first, ArcLight's fluorescence follows the leftward Q-V shift in gating current that is a hallmark of the relaxed state in Ci-VSP (*Villalba-Galea et al., 2008*) (*Figure 7—figure supplement 2*). Second, ArcLight's deceleration of off kinetics increases as the duration of the depolarizing pulse increases (*Figure 7—figure supplement 2*), as occurs in the relaxation process of voltage-gated potassium channels (*Lacroix et al., 2011*). Thus, exit of the VSD from the relaxed state may be linked to the extremely slow fluorescence kinetics. Finally, VSD relaxation explains the comparatively flat kinetics observed in ArcLight gating currents during positive holding potentials. Since these holding potentials dramatically left-shift the $V_{1/2}$ of the ArcLight Q-V to values well outside experimentally-accessible values (*Figure 7—figure supplement 2*), the corresponding observed τ-V curve (*Figure 4F*) represents only a single tail of the true τ-V, and thus appears to be relatively constant.

ArcLightning is unique when compared to many previously developed fluorescent probes. Its speed is not as rapid as ElectricPk (*Barnett et al., 2012*), but it has a much larger voltage-sensitive fluorescence response. As opposed to ArcLightning, both ArcLight and the more recent voltage indicator ASAP1 have significant slow components in their off response (*Jin et al., 2012*; *St-Pierre et al., 2014*). Accordingly, further study of the interactions between VSD movement and fluorescence response, including the contribution of the relaxed state, of voltage-sensitive phosphatase derived GEVIs should prove useful (*Villalba-Galea et al., 2009*). Improved understanding of this mechanism may lead to additional probes that accomplish what ArcLightning accomplishes: rapid kinetic responses in response to hyperpolarization that prevent the drifting baselines observed in response to repetitive pulses in similar probes (*Piao et al., 2015*) and a roughly linear voltage-sensitivity across the physiological voltage range that could allow for the visualization of both subthreshold and action potential activity. FRET-based voltage indicators, including VSFP, Mermaid, and Butterfly indicators have recently achieved many or all of these characteristics (*Knöpfel et al., 2015*). Many of these probes have slightly larger fluorescence changes than ArcLightning and comparable kinetics, and the recently developed chimeric VSFP Butterflies appear particularly promising (*Mishina et al., 2014*). However, ArcLightning and other monochromatic GEVIs have the distinct advantage of freeing other spectral channels for use in measuring other parameters, such as cell morphology or changes in calcium concentration. Additionally, as ArcLighting requires the recording of only one fluorophore's emission, it is more compatible with simpler optical setups. In addition to the ability demonstrated here to accelerate the voltage-sensitive response of ArcLight, the previously reported I126F mutation (*Lacroix and Bezanilla 2012*) may also be applicable to other GEVIs such as ASAP1 (*St-Pierre et al., 2014*) and the Butterfly indicators (*Akemann et al., 2015*). Finally, the single-molecule and macroscopic fluorimetric techniques presented here will improve our future capabilities to mechanistically understand VSP-derived GEVIs and to engineer new and improved generations of these probes.

# Materials and methods

## Molecular biology

ArcLight for *Xenopus laevis* expression was created in the lab by fusing Venus to Ci-VSP in the SP64T plasmid vector at the appropriate location and mutating to the super ecliptic pHluorin A227D to generate ArcLight or ArcLight' (*Jin et al., 2012*). ArcLight' was identical to ArcLight but with two point mutations (L64F and T65S) which return the fluorophore to the wild-type GFP scaffold rather than the eGFP scaffold; thus, ArcLight' contains the ecliptic pHluroin A227D fluorophore rather than the super ecliptic pHluorin A227D fluorophore. The two constructs were shown to have nearly identical voltage-dependence (*Jin et al., 2012*), and in our hands ArcLight' appeared to perform better in single-molecule experiments. ArcLight' GFP domains and eGFP were expressed in *E. Coli* in the pQE-32 plasmid vector using previously-described methods (*Negro et al., 1997*). ArcLight for mammalian expression was a gift from Vincent Pieribone (Addgene plasmid #36856) (*Jin et al., 2012*). Mutations I126F and Q217R were generated in these constructs using site-directed mutagenesis by polymerase chain reaction and subsequently verified by sequencing. For oocyte expression, DNA was prepared using the NucleoSpin Plasmid kit (Macherey-Nagel, Bethlehem, PA) and linearized with NotI (New England Biolabs, Ipswich, MA). Linearized cDNA was transcribed to RNA with the mMESSAGE mMACHINE Sp6 kit (Life Technologies, Carlsbad, CA). Oocytes were injected with either 0.25 to 1 ng of RNA (single-molecule recordings) or 50 ng of RNA (macroscopic recordings) and incubated at 16°C in solution containing (in mM) 96 NaCl, 2 KCl, 1.8 CaCl$_2$, 1 MgCl$_2$, 10 HEPES, at pH 7.4 with 10 mg/L of gentamicin. Recordings were made 1–4 days following injection. For mammalian cell expression, DNA was prepared using the NucleoBond Xtra Midi Plus kit (Macherey-Nagel). DNA was then transfected into HEK293 cells that had been previously plated on glass coverslips at low density, using Lipofectamine LTX reagent (Thermo Fisher Scientific, Waltham, MA). HEK cells were incubated at 37°C with 5% CO$_2$ in DMEM medium with HEPES and no phenol red (Thermo Fisher Scientific) for three to four days prior to recording.

## Single-molecule fluorescence

Glass slides were prepared ahead of time by incubation in piranha solution (70% H$_2$SO$_4$ at 18M / 30% H$_2$O$_2$ at 30% w/w, both from Sigma-Aldrich, St. Louis, MO) followed by copious rinsing with water and storage under water in a 50 mL tube until use. When ready to use, a slide was removed and dried under a stream of nitrogen, and a chamber cut from cured Sylgard 184 (Dow Corning, Midland, MI) was placed on the dry slide. A solution of 200 mM polyethylenimine (Mw ≈ 750,000, Sigma-Aldrich) in SOS (96 mM NaCl, 2 mM KCl, 1 mM MgCl$_2$, 1.8 mM CaCl$_2$, 20 mM HEPES) was placed into the chamber and left for 30 minutes, followed by extensive rinsing with fresh SOS. This recording chamber was filled with SOS for recording. To allow TIRF microscopy, oocytes were placed in a high osmolality 'shrinking' solution prior to recording and the vitelline membranes were mechanically removed (*Sonnleitner et al., 2002*). Oocytes were then placed into the recording chamber and mounted on the microscope.

ArcLight fluorescence from single molecules was recorded by an Evolve 128 EMCCD camera (Photometrics, Tucson, Arizona) attached to a home-built TIRF setup based on an Olympus IX71 inverted microscope (Center Valley, Pennsylvania) with a 60X/1.45 NA microscope objective (Olympus). Excitation was provided by a 473 nm DPSS laser (Shanghai Dream Lasers Technology Co., Ltd., Shanghai, China). Typical excitation intensities were around 150 W/cm$^2$. Fluorescence was observed through a T495lpxt dichroic and ET500lp emission filter (Chroma Technology Corp., Bellows Falls, VT). Voltage clamp was performed in a two-electrode configuration with a Warner Instruments OC-725A amplifier (Hamden, Connecticut). Both the electrophysiological and optical equipment were controlled using in-house software. For single-molecule recordings, the recording chambers were cooled to approximately 14°C by cooling the microscope objective lens assembly with recirculating chilled water.

The bacterially-expressed ArcLight' GFP domain and eGFP were incubated on PEGylated glass slides possessing a low density of Cu-NTA chelates (Cu_01, MicroSurfaces, Inc., Englewood, NJ) as described previously (*Holtz et al., 2007*). Since the proteins contained poly-histidine tags, they efficiently bound the copper chelate while a subsequent rinse with fresh buffer removed excess

proteins. As this point, the slides were imaged using TIRFM with the same optical configuration as described above.

## Macroscopic recordings

Simultaneous recordings of ArcLight gating currents and fluorescence responses were performed using the cut-open oocyte voltage-clamp technique (*Stefani and Bezanilla, 1998*) in combination with a photodiode to measure temporal changes in fluorescence emission (*Cha and Bezanilla, 1998*). Gpatch, an in-house program, controlled an SB6711 digital signal processor-based board (Innovative Integration, Simi Valley, CA) with an A4D4 conversion board (Innovative Integration, Simi Valley, CA). Oocytes were held under voltage-clamp with a Dagan CA-–1B amplifier (Minneapolis, MN) and filtered at 2–5 kHz depending on the sampling rate. ArcLight emission fluorescence was collected through an Olympus LUMPlan FL N 40X/0.8 NA water-immersion objective by a PIN-020A photodiode (UDT Technologies, Torrance, CA), amplified by a patch clamp amplifier L/M-EPC-7 by LIST Medical Electronic (Darmstadt, West Germany) with a filter of 10 kHz, and then integrated over each sampling period using a home-built integrator circuit. ArcLight was excited via a ThorLabs LED controller triggering a 470 nm LED (ThorLabs, Newton, New Jersey) that was passed through a filter cube housing a 480/40 excitation filter, a 505 long-pass dichroic, and a 535/50 emission filter (Chroma, Bellows Falls, VT). All recordings were performed at around 19°C, with an external solution containing (in mM) 120 N-methyl-D-glucamine/methanesulfonic acid (NMG/MES), 10 HEPES, and 2 $Ca(OH)_2$ and an internal solution containing (in mM) 120 NMG-MES, 10 HEPES, and 2 EGTA. Both solutions were set to pH 7.5. Current microelectrodes pulled on a Flaming/Brown micropipette puller (Sutter Instruments, Novato, CA, model P-87) were filled with 3 M KCl and had a resistance of ~0.2–0.9 M.

## GEVI characterization

HEK cells were patch-clamped using an Axon Instruments Axopatch 200A amplifier (Molecular Devices, Sunnyvale, CA). Signals were filtered (Frequency Devices, Ottawa, Illinois, model 950L8L) and digitized with an SBC-6711-A4D4 data acquisition board (Innovative Integration). Patch pipettes with resistances of roughly 5 MΩ were pulled on a $CO_2$ laser micropipette puller (Sutter Instruments, model P-2000). The bath temperature was raised and maintained using a home-built system with an Omega CNPt series temperature controller (Stamford, CT).

## Data analysis

Data analysis was performed in MATLAB (The MathWorks, Inc., Natick, MA), as well as in-house software. Weighted time constants ($\tau_w$) of gating current were taken using double exponential fits to the decay phase of the gating current. Kinetics of fluorescence traces were first taken from double exponential fits to the trace; when fit with two exponentials, fluorescence traces obtained from off pulses in oocytes only had one meaningful time constant, and thus were refit with one exponential. For clarity of comparison, all kinetics were reported as $\tau_1$, which denotes the faster and larger amplitude component of the fluorescence traces. Q-V curves were calculated by normalizing the integral of the gating current following a sloped baseline subtraction to remove leak current. When holding at +40 mV the Q-V and F-V curves did not approach saturation in the hyperpolarized direction. Therefore, we first normalized the Q-V by the total gating charge moved in the same oocyte during a depolarizing pulse from a holding potential of -120 mV as these traces saturate on both ends and total gating charge is conserved (*Villalba-Galea et al., 2008*). The normalization factor found in the Q-V was used to normalize the F-V when holding at +40 mV.

## Acknowledgments

Thanks to Dr. Tomoya Kubota for generation of the ArcLight' construct for oocyte expression, Dr. João de Souza for patch-clamp assistance, and Josh Aaker and Dr. Brian Popko for HEK cells and transfection instruction.

## Additional information

### Funding

| Funder | Grant reference number | Author |
|---|---|---|
| National Institute of General Medical Sciences | R01-GM030376 | Francisco Bezanilla |
| National Institute of General Medical Sciences | T32-GM07281 | Jeremy S Treger |
| National Institute of Neurological Disorders and Stroke | F31-NS081954 | Michael F Priest |

The funders had no role in study design, data collection and interpretation, or the decision to submit the work for publication.

### Author contributions

JST, MFP, Conception and design, Acquisition of data, Analysis and interpretation of data, Drafting or revising the article; FB, Conception and design, Analysis and interpretation of data, Drafting or revising the article, Contributed unpublished essential data or reagents

### Ethics

Animal experimentation: All animals were treated in accordance with protocols approved by the University of Chicago Animal Care and Use Committee.The frog protocol number is ACUP 71475.

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
