## [Decision Letter]

Thank you for submitting your work entitled "Single-molecule fluorimetry and gating currents inspire an improved optical voltage indicator" for peer review at *eLife*. Your submission has been favorably evaluated by Gary Westbrook (Senior Editor) and three reviewers, one of whom, Richard Aldrich, is a member of our Board of Reviewing Editors. The other two reviewers, Brian Salzberg and Kenton Swartz, also agreed to share their identities.

The reviewers have discussed the reviews with one another and the Reviewing editor has drafted this decision to help you prepare a revised submission.

This is an excellent manuscript that studies the properties of the genetically encoded voltage indicator (GEVI) ArcLight, together with those of ArcLightning, a novel GEVI introduced by the authors in this paper. The authors use the ArcLight GEVI to track voltage-sensing domains (VSDs) at the single molecule level, a not inconsiderable achievement. They also characterize, very carefully, the kinetics of ArcLight's fluorescence changes, and identify a previously unrecognized (or, at least, unreported) delay. As a result of their studies, they develop ArcLightning, a GEVI with superior kinetics, and they posit a four-state minimal model for fluorescence changes of GEVIs. This manuscript provides both a novel, and potentially extremely useful new GEVI, as well as important insights into the behavior of genetically expresses voltage-indicators. It also shows, as the authors note, that mutations found during biophysical studies of voltage sensor behavior can inform the rational design of new and improved GEVIs. Also included is an exciting, however preliminary, description of single-molecule fluorescence signals form single voltage sensors.

We are enthusiastic about the work on the properties and modification of ArcLight, and believe that necessary revisions should be easily accomplished. However, we feel that the single-molecule work, while potentially quite exciting, is much too preliminary for inclusion, and should be removed. We would be quite enthusiastic about receiving an additional paper describing a thorough and rigorous characterization of the single voltage sensor signals.

Essential revisions:

1) Remove the single-molecule studies from the paper.

2) The authors demonstrate a close correlation between Q and F changes for the wt S4 construct, as well as the R217Q and R217E constructs, which is great to see, but then don't carefully analyze (or at least present) the multiexponential kinetics of either measure and look to see which match and which don't. It would be nice to show integrated Q traces, analyze those, and compare them with what is seen for F. With the two other mutants, it would be interesting to see how these change. The lag in F is also interesting, and it would be nice to show measure V traces as well as integrated Q traces so the reader can see how both V and Q change relative to F.

3) Please provide a comparison of the advantages and disadvantages of ArcLightning with other versions of optical voltage sensors (e.g. Mermaid, ElectricPk, VSFP).

---

## [Author Response]

Essential revisions: 1) Remove the single-molecule studies from the paper.

Thank you for this comment, and we agree that it is too early to draw conclusions about voltage sensor function based on the data presented. Accordingly, in consultation with the editors, we have substantially revised the presentation of the single-molecule work. We have eliminated entirely the discussion of using ArcLight to learn about voltage sensor function. However, we strongly feel that the fluctuation noise discovered here has substantial implications for future GEVI use and development. In particular, the macroscopic summation of the noise, as well as the corresponding decrease in effective quantum yield compared to a non-fluctuating fluorophore, very likely serve to decrease the signal-to-noise ratio of many GEVI-derived signals, thus decreasing the resolving power of these tools for small signals. In addition to this, correctly-modulated pulsing may be of benefit for increasing signal from many GFP-based GEVIs, as previous work has demonstrated that this can improve signal form many fluorophores that visit dark states. Thus the discovery of this noise may serve to motivate new avenues of GEVI optimization that are distinct from and complementary to current directions. We hope you agree with us that this describes an important phenomenon that likely afflicts many current GEVIs and that it should remain a part of this manuscript in its revised form.

2) The authors demonstrate a close correlation between Q and F changes for the wt S4 construct, as well as the R217Q and R217E constructs, which is great to see, but then don't carefully analyze (or at least present) the multiexponential kinetics of either measure and look to see which match and which don't. It would be nice to show integrated Q traces, analyze those, and compare them with what is seen for F. With the two other mutants, it would be interesting to see how these change. The lag in F is also interesting, and it would be nice to show measure V traces as well as integrated Q traces so the reader can see how both V and Q change relative to F.

Thank you for these insightful suggestions. We have added two new figures to the section titled “ArcLight Fluorescence Changes are Slower than Gating Currents and Possess a Lag” that address these points. First, we compare the two time constants of both integrated gating charge and fluorescence and show that there are no obvious kinetic correlations between the two. We then show a pair of representative examples of overlaid V, Q, and F traces to more clearly show how they change in time relative to each other. Finally, we do a kinetics analysis of gating currents, fluorescence lags, and fluorescence kinetics for the R217R and R217E constructs and show that they have the same general behavior as the wild-type construct.

3) Please provide a comparison of the advantages and disadvantages of ArcLightning with other versions of optical voltage sensors (e.g. Mermaid, ElectricPk, VSFP).

The toolbox available to researchers interested in measuring changes in membrane potential grows larger each year and we are happy to compare our tool to those currently available. Additional information has been added in the last paragraph of the Discussion. Briefly, ElectricPk is faster but much smaller, and Mermaid and VSFP (and other closely related FRET based probes) are comparable in their abilities but take up a larger fraction of the visible light spectrum, making optical measurement of additional parameters challenging.